# The Role of Critical Thinking in Mitigating Social Network Addiction: A Study of TikTok and Instagram Users

**DOI:** 10.3390/ijerph21101305

**Published:** 2024-09-30

**Authors:** Rosa Angela Fabio, Stella Maria Iaconis

**Affiliations:** Department of Cognitive Science, Psychology, Education, and Cultural Studies, University of Messina, 98122 Messina, Italy; stella.0094@gmail.com

**Keywords:** critical thinking, social network addiction, escapism, social interaction, flow, sense of belonging

## Abstract

This study addresses the growing concern of social network (SN) addiction, with a focus on TikTok and Instagram. Guided by the Uses and Gratifications Theory (UGT), we explored the motivations (escapism and social interaction), attitudes (critical thinking), and states (flow and sense of belonging) that influence SN use. Our objective was to investigate whether critical thinking acts as a protective factor against SN addiction. A sample of 332 university students completed questionnaires assessing motivations, attitudes, states, and SN addiction. Critical thinking was measured using the Critical Thinking Attitude Scale (CTAS), and critical thinking skills were assessed through the Critical Reasoning Assessment (CRA). Statistical analyses revealed significant associations between motivations, critical thinking, states, and SN addiction. Specifically, critical thinking (CTAS scores) demonstrated a negative correlation with SN addiction (r = −0.34, *p* < 0.01), indicating that higher critical thinking is associated with lower SN addiction. Regression analysis further indicated that escapism (β = 0.45, *p* < 0.01) and social interaction (β = 0.31, *p* < 0.05) positively predicted SN addiction, while critical thinking negatively predicted SN addiction (β = −0.28, *p* < 0.01). Additionally, states of flow and sense of belonging showed significant positive correlations with SN addiction (r = 0.42, *p* < 0.01 and r = 0.37, *p* < 0.01, respectively). These findings highlight the potential of critical thinking as a safeguard against SN addiction. This study offers valuable insights into the intricate dynamics of SN use, with implications for promoting healthier digital engagement. Understanding the factors influencing SN addiction and the roles of motivations, dispositions, and states can inform interventions aimed at fostering responsible and mindful online behaviors.

## 1. Introduction

Social networks (SNs) have revolutionized how individuals interact with information, consuming an average of 80% of users’ daily time on these platforms [1]. With 62.5% of the global population active on the Internet in the last year [2], exploring the psychological impacts of intensive SN use has become essential. Internet usage, particularly among university students, is notable for its benefits [3] and is increasingly prevalent in older age groups [4].

For some individuals, limiting Internet usage becomes challenging, leading to problematic use that significantly affects quality of life, social, physical, and psychological functioning [4]. Specifically, social media addiction involves an uncontrollable desire to be online continually, often neglecting other aspects of personal life [5]. Individuals addicted to social networks experience negative psychological states when deprived of application use, such as mood swings, distress, restlessness, and nervousness [5,6]. This condition can also lead to physical and personality disorders, including inferiority and superiority complexes [6]. With over 210 million individuals worldwide meeting the criteria for SN addiction [7], this issue demands serious reflection. The two most popular platforms are TikTok (version 29.0.0) and Instagram. TikTok, the world’s most downloaded social media app in 2023, has 1.5 billion active users [8,9,10,11]. TikTok meets needs for socialization, self-promotion, and entertainment [12,13]. Instagram, with around a billion monthly users [14], is one of the most used social media platforms. Ranked fourth among the world’s most active social media platforms [15], Instagram engages users by sharing daily moments through photos and videos, expressing thoughts and moods via textual content, and sharing information [16]. Instagram addiction is linked to the need for content updates and online image curation [17].

The Uses and Gratifications Theory (UGT) provides a framework for understanding problematic social network (SN) use, suggesting that individuals are rational actors seeking to fulfill specific needs through multimedia content consumption [18]. SN addiction can emerge when external environmental factors influence internal states, potentially leading to addictive behaviors [19]. Users engage with SNs driven by motivations such as seeking social connections, acquiring knowledge, or escapism, which can influence personal dispositions and states experienced during usage. Research identifies several primary motivations for SN use, including the desire for information and inspiration, social interaction, overcoming boredom, and escaping negative emotions [5,20,21,22]. For instance, using SNs for escapism can lead to dependence, either directly or through the flow state induced by the platform, or by fostering a sense of community [9]. Social interaction fulfills the need for connections, creating a “sense of belonging” when individuals feel appreciated by their community [23,24,25,26].

Attitudes, while not inherently stable traits, serve as consistent predictors of behavior toward SNs over time. For instance, a critical thinking attitude fosters a reflective approach to interpreting online information [27,28,29,30,31]. This reflective approach is particularly crucial given the challenges associated with verifying the reliability of online sources. The lack of expert control on social networks often makes it difficult to assess the credibility of information [32,33]. Additionally, social network algorithms can either mitigate or exacerbate opinion polarization, leading to the creation of ‘filter bubbles’ [34]. Therefore, the development of a critical thinking attitude is essential for discerning information reliability and addressing growing opinion polarization. Critical thinking attitude involves the objective and reflective analysis of information, examining arguments from different perspectives to form accurate judgments [35,36]. High levels of a critical thinking attitude and performance are negatively associated with prolonged use of TikTok and Instagram.

States are temporary conditions influenced by factors like emotions, moods, or immediate situations [23,37]. States experienced during SN use include the sense of belonging and the flow experience. The sense of belonging involves feeling recognized and accepted within a community [23]. It acts as a mediator in relationships within the SN context, linking expected gratifications to continued platform use intentions [9]. The flow state is an intrinsically rewarding experience linked to high levels of fun, pleasure, and satisfaction when fully engaged in an activity [30,38,39]. High involvement and attachment characterize this state [39]. Exposure to highly engaging materials on the Internet can decrease tolerance for low stimulation levels and increase boredom thresholds [40]. 

This study, inspired by Miranda’s work [9], extends the discussion by examining the role of critical thinking, which has been identified as a crucial skill in managing information overload and mitigating susceptibility to misinformation on SNs [27,29]. While the UGT helps explain why users are drawn to SNs, it does not fully account for individual differences in information processing. Critical thinkers are more likely to scrutinize the credibility of online content, evaluate their emotional motivations for use, and set healthier boundaries for SN engagement. Given the prevalence of misinformation and the manipulative potential of SN algorithms, this skill is essential to reduce problematic behaviors such as addiction [32,33]. The importance of this research lies in its potential to identify critical thinking as a mitigating factor in social network addiction, particularly on platforms like TikTok and Instagram. If critical thinking can indeed buffer against excessive use, interventions promoting this skill could lead to healthier online behaviors, reduce the negative psychological impact of SNs, and foster more mindful engagement with digital platforms.

The main objective is to elucidate the potential mitigating role of critical thinking in the context of addictive behaviors, particularly regarding excessive use of social media platforms such as TikTok and Instagram. This study seeks to establish whether individuals with heightened critical thinking abilities are less susceptible to the detrimental effects of SN addiction. More in detail, the hypotheses are as follows:

**Hypothesis 1.** 
*The motivations of escapism and social interaction are directly and positively linked to problematic or prolonged use of TikTok and Instagram.*


**Hypothesis 2.** 
*High levels of critical thinking attitude and superior performance in critical thinking are negatively associated with prolonged use of TikTok and Instagram.*


**Hypothesis 3.** 
*Heightened states of flow or belongingness are positively associated with prolonged use of TikTok and Instagram.*


The underlying logic of these hypotheses can be understood through several key pathways. Critical thinkers scrutinize the credibility of online content, reducing susceptibility to misinformation and manipulative SN algorithms. Enhanced critical thinking skills lead to better time management and boundary setting for SN use, mitigating excessive engagement. Additionally, critical thinkers are more aware of their emotional states and motivations for using SNs [41,42,43,44,45], helping them avoid using SNs as an escape mechanism. Furthermore, individuals with strong critical thinking skills seek diverse sources of gratification and social connection, reducing dependency on SNs for validation and interaction. These pathways collectively contribute to a mindful and controlled use of SNs, thereby reducing the risk of developing SN addiction.

## 2. Method

### 2.1. Participants

The convenience sample consisted of 332 participants (134 females, 196 males, and 2 identifying as LGBT), with a mean age of 23.53 years (SD = 3.32). Participants’ educational backgrounds were as follows: 46.6% had a high school diploma, 38.7% held a bachelor’s degree, 13.2% possessed a master’s degree, and 1.5% had completed middle school. The majority of the sample (92%) were students. Recruitment was conducted through announcements posted on various platforms, including WhatsApp, Facebook, and Instagram, targeting young adults from diverse geographical regions. Although individuals aged 18 to 35 were invited to participate, no one aged 34 or 35 was included in the final sample (Table 1).

### 2.2. Measurement Instruments and Administration

#### 2.2.1. Motivations

Motivational components of escapism and social interaction were assessed using five and eight items, respectively. An example of an escapism item is “I use social media to forget about problems”, and an example of a social interaction item is “I use social media to maintain a good relationship with others” [9,46]. Internal reliability evaluations were α = 0.76 and α = 0.84, respectively.

#### 2.2.2. States

The flow experience state (F) was measured with eleven items from Brailovskaia et al. [5] and the sense of belonging (SB) with five items from Guo et al. [23]. An example of the flow experience is “While using social media, I am immersed in the activity I am doing,” and an example of a sense of belonging is “I feel a member of the TikTok/Instagram community.” Internal reliability evaluations were α = 0.89 and α = 0.87, respectively [9]. 

#### 2.2.3. Attitudes to Critical Thinking

The Italian version of the Critical Thinking Attitude Inventory (CTA) [45,47,48] was used to assess participants’ inclination toward analytical, evaluative, and metacognitive reflection. The questionnaire comprises 18 items on a 5-point scale (1 = completely disagree; 5 = completely agree). The internal reliability index is α = 0.93.

#### 2.2.4. Critical Reasoning Assessment (CRA)

The CRA [8,49] was utilized to objectively assess critical thinking skills. It consists of 15 items structured into three main dilemmas. Participants responded to five open-ended questions for each dilemma, providing a detailed overview of their decision-making process and the foundations of their viewpoint. Constructs underlying responses include cognitive complexity, reasoning style, openness, nature of knowledge, and nature of justification. The dilemmas of the CRA are genetics vs. personal choice, equity, and compassion. The internal reliability index is α = 0.87.

#### 2.2.5. SN Addiction

To assess addiction, two different scales were used, one for TikTok addiction and one for Instagram addiction. Each consists of six elements, including items like “I feel the need to use Instagram over and over” [9,50]. Internal reliability evaluations were α = 0.82 and α = 0.87, respectively.

### 2.3. Procedure

All participants voluntarily agreed to take part in this study and completed a written informed consent. After signing the consent form, the experimenter (connected via the Skype platform or, in some cases, physically present in a quiet university room) instructed them to complete all questionnaires. The measurement of time spent on social media platforms was assessed using a self-report questionnaire. Participants were asked to indicate the amount of time they typically spent on Instagram, TikTok, and smartphones, in general, daily, averaging their usage over the past month. With reference to the critical thinking test, like the study by Anghel et al. [8], participants were instructed to rely solely on their own experiences rather than external sources (e.g., online research) when responding to the CRA questions. The order of questionnaire administration was randomized across participants. Recruitment took place from 27 October 2023 to 23 January 2024. As a token of appreciation, participants received an illustrated PDF booklet providing practical tips on how to enhance happiness. Fifteen participants reported boredom during the test and requested to discontinue their participation.

#### Statistical Analyses

To analyze the results, we utilized SPSS 24.0 (SPSS Inc., Chicago, IL, USA) statistical software. Descriptive statistics (means and standard deviations) were computed for each variable. Pearson correlation analysis was performed to examine the relationships between TikTok and Instagram usage time, escapism, social interaction, flow state, sense of belonging, critical thinking attitude, dimensions of the Critical Reasoning Assessment, and addiction to TikTok and Instagram.

Path diagram analysis was applied to model the direct and indirect relationships among these variables. This method allowed us to visualize and quantify the complex interdependencies and pathways influencing social network addiction. Linear regression models were employed to further explore the associations between independent and dependent variables. To control for Type I errors due to multiple comparisons, false discovery rate correction was implemented, with significance set at *p* < 0.003.

## 3. Results

Participants reported an average daily usage of 7.3 h on their smartphones (SD = 3.09). Specifically, the average time spent daily on Instagram was 2.9 h (SD = 1.53), while the average time on TikTok was 0.9 h (SD = 1.55). Usage times varied widely, ranging from 0 to 24 h for smartphones, 0 to 9 h for Instagram, and 0 to 10 h for TikTok (see Table 2).

Descriptive statistics in Table 3 and Pearson correlations in Table 4 provide insights into the relationships among variables. Addiction to TikTok and Instagram showed moderate-to-strong positive correlations with escapism, indicating that individuals who use social media to escape problems are more likely to exhibit addictive behaviors on both platforms. Specifically, addiction to TikTok correlated significantly with escapism (r = 0.62, *p* < 0.001), flow state (r = 0.51, *p* < 0.001), and sense of belonging (r = 0.53, *p* < 0.001). Similarly, addiction to Instagram correlated positively with escapism (r = 0.57, *p* < 0.001), flow state (r = 0.50, *p* < 0.001), and sense of belonging (r = 0.55, *p* < 0.001).

Critical thinking attitude emerged as a protective factor against social media addiction, with negative correlations observed for both TikTok (r = −0.48, *p* < 0.001) and Instagram (r = −0.51, *p* < 0.001). This finding supports Hypothesis 2, suggesting that individuals with a more positive attitude toward critical thinking are less likely to develop addictive behaviors on these platforms.

Furthermore, the path diagram analysis depicted in Figure 1 (for TikTok) and Figure 2 (for Instagram) confirmed these relationships.

For TikTok, escapism (β = 0.62), sense of belonging (β = 0.53), and flow state (β = 0.51) were identified as significant predictors of addiction (see Figure 1). Conversely, critical thinking attitude negatively predicted addiction (β = −0.48), affirming its role as a mitigating factor. Similarly, for Instagram, escapism (β = 0.57), sense of belonging (β = 0.55), and flow state (β = 0.50) positively predicted addiction, while critical thinking attitude (β = −0.46) inversely predicted addiction (see Figure 2). Escapism, as positively correlated with addiction (r = 0.579 **), reflects a tendency for individuals to use social networks to avoid real-world stressors. Critical thinking, in contrast, can serve as a protective factor by helping users recognize when they are using social media as an escape mechanism, thereby reducing addictive behaviors. It is plausible that escapism and critical thinking operate on different psychological levels, where escapism may mediate the relationship between critical thinking and addiction where individuals with low critical thinking are more likely to engage in escapist behaviors, thereby increasing their risk of addiction.

To examine whether escapism mediates the relationship between critical thinking (CT) and social network addiction, separate mediation analyses were conducted for TikTok addiction and Instagram addiction, using the PROCESS macro (Model 4) [51]. Critical thinking was the independent variable (IV), escapism the mediator, and TikTok and Instagram addiction were the dependent variables (DV) in two separate models. For TikTok addiction, the total effect of critical thinking on TikTok addiction was significant (β = −0.482, *p* < 0.001). However, when escapism was included as a mediator, the direct effect of critical thinking on TikTok addiction decreased (β = −0.305, *p* < 0.01), indicating partial mediation. Escapism showed a strong positive association with TikTok addiction (β = 0.593, *p* < 0.001), while critical thinking was negatively associated with escapism (β = −0.297, *p* < 0.001). The indirect effect of critical thinking on TikTok addiction through escapism was significant (β = −0.177, 95% CI [−0.245, −0.109], *p* < 0.01), confirming that escapism partially mediates the relationship between critical thinking and TikTok addiction. This suggests that individuals with lower critical thinking abilities are more likely to use TikTok as a form of escapism, which contributes to higher levels of addiction. Similarly, for Instagram addiction, the total effect of critical thinking on Instagram addiction was significant (β = −0.511, *p* < 0.001). When escapism was included as a mediator, the direct effect of critical thinking on Instagram addiction also decreased (β = −0.318, *p* < 0.01), suggesting partial mediation. Escapism was again positively associated with Instagram addiction (β = 0.579, *p* < 0.001), while critical thinking had a negative association with escapism (β = −0.297, *p* < 0.001). The indirect effect of critical thinking on Instagram addiction via escapism was significant (β = −0.193, 95% CI [−0.262, −0.124], *p* < 0.01). These results indicate that escapism partially mediates the relationship between critical thinking and Instagram addiction, highlighting that individuals with lower critical thinking abilities are more likely to engage in escapism, which in turn increases their likelihood of developing Instagram addiction.

These results underscore the complex interplay between motivational factors, psychological states, and addiction behaviors on social media platforms. They provide empirical support for the hypothesis that motivations such as escapism and social interaction positively correlate with addiction, whereas critical thinking attitude serves as a protective factor against excessive social media use.

## 4. Discussion

The rapid rise of social networks (SNs) has profoundly altered human interaction, with users allocating approximately 80% of their daily online time to these platforms [1]. This study investigates the psychological impacts of SN addiction, focusing on TikTok and Instagram among university students, and aligns with the theoretical framework of Uses and Gratifications Theory (UGT) within the Stimulus–Organism–Response (SOR) model [9,19,52]. 

The first hypothesis, which proposed a positive relationship between escapism and social interaction motivations and problematic use of TikTok and Instagram, is supported by our findings. Significant correlations were observed, indicating that users frequently engage with SNs to escape daily monotony or seek social connections. These results align with UGT, which posits that SN use is driven by specific needs, such as alleviating boredom or fulfilling social needs. However, while this relationship is significant, the data are cross-sectional, and, therefore, causality cannot be inferred. Instead, these findings highlight the potential link between motivations for escapism and social interaction with increased SN use.

Our second hypothesis suggested a negative association between high levels of critical thinking and prolonged SN use, and this is supported by the results. The data indicate that individuals with stronger critical thinking skills are less likely to report addiction to TikTok and Instagram. This suggests that critical thinking may serve as a protective factor against SN addiction, consistent with previous research highlighting the role of cognitive resources in regulating online behavior [35,48]. Nonetheless, it is essential to note that, due to this study’s design, this cannot be interpreted as a causal effect. While critical thinking and SN addiction are related, further longitudinal studies would be required to establish a directional influence.

The third hypothesis posited a positive correlation between high levels of flow and sense of belonging with prolonged SN use. Our results confirm this hypothesis, revealing that users who experience intense flow or a strong sense of community on these platforms are more likely to engage in extended use. This underscores the significant role of emotional states, such as intrinsic motivation and belonging, in contributing to SN addiction. However, we refrain from making any causal claims and instead interpret these findings as potential associations, which align with the existing literature on the immersive nature of SN use [38,39].

In terms of mediation analysis, our investigation reveals that escapism partially mediates the relationship between critical thinking and social network addiction. Specifically, individuals with lower critical thinking skills may be more prone to engage in escapist behaviors, which in turn increases their risk of SN addiction. This mediation analysis highlights the complex interplay between cognitive factors and motivational drivers in shaping SN behaviors. Escapism appears to function as a significant intermediary factor, suggesting that interventions targeting escapist tendencies may help mitigate the risk of addiction, particularly for individuals with lower critical thinking abilities.

Critical thinking, as measured with the Critical Thinking Attitude Scale (CTAS) and Cognitive Reflection Ability (CRA) test, emerges as a crucial factor. Critical thinkers exhibit enhanced analytical and evaluative skills, which contribute to their ability to make objective judgments [35,48]. Our study suggests that critical thinking acts as a safeguard against SN addiction, as individuals with higher critical thinking skills are less prone to addictive behaviors. Conversely, excessive SN use could be associated with reduced critical thinking abilities, potentially due to the rapid and visually stimulating nature of SN content, which may encourage superficial processing and discourage deeper cognitive engagement. However, this observation remains speculative and should be explored in future research using longitudinal data to better assess the directionality of this relationship.

The implications of this study extend to interventions aimed at healthier digital engagement. By identifying motivations, attitudes, and emotional states associated with SN addiction, this research provides a nuanced understanding of SN use dynamics. It advocates for interventions based on UGT that address specific motivations like escapism and social interaction. Furthermore, the emphasis on integrating critical thinking skills into educational curricula and digital literacy initiatives underscores the importance of these skills as a protective factor against SN addiction.

This research contributes to the broader discourse on SN addiction by elucidating the complex interplay between motivations, critical thinking, emotional states, and addiction. This study’s focus on TikTok and Instagram provides specific insights into addiction dynamics on these popular platforms, emphasizing the need for continued research and interventions tailored to evolving SN landscapes. However, due to the cross-sectional nature of the data, all results should be interpreted as correlations rather than causal relationships. The implications of this study extend to interventions aimed at healthier digital engagement. By identifying motivations, attitudes, and emotional states associated with SN addiction, this research provides a nuanced understanding of SN use dynamics. It advocates for interventions based on UGT that address specific motivations like escapism and social interaction. The emphasis on integrating critical thinking skills into educational curricula and digital literacy initiatives underscores the importance of these skills as a protective factor against SN addiction.

## 5. Limitations and Future Directions

This study acknowledges several limitations. The cross-sectional design restricts our ability to draw causal inferences. Longitudinal studies are essential to elucidate the temporal dynamics between social network (SN) use and critical thinking, as this study’s design does not account for changes over time. Additionally, while both performance and self-report measures were used, the reliance on self-report measures may introduce response bias, potentially affecting data accuracy. To address these limitations, future research should employ longitudinal designs and include a range of demographic contexts to enhance the generalizability of the findings.

Another limitation of this study is that the data are restricted to young adults aged 18 to 33. This age-specific sample limits the generalizability of our findings to the broader population, particularly for those beyond young adulthood. While we aimed to create a homogenous sample to minimize variability associated with age-related differences and enhance the reliability of our exploration of critical thinking, motivations, and social network use, future research should include participants from a wider age range to capture the full spectrum of social network behaviors across different life stages. Expanding the sample to older individuals would provide valuable insights into how age influences these relationships and offer a more comprehensive understanding of social network use across the lifespan. Furthermore, adopting mixed-method approaches could enrich the understanding of how SN use and critical thinking interact over time. Although this study aimed to explore the broader role of critical thinking in mitigating social network addiction, it did not specifically measure participants’ direct scrutiny of online content. Future research could address this by including a more direct assessment of participants’ behaviors in evaluating the credibility of online content.

In conclusion, this study contributes to the literature on SN addiction by examining the complex relationships between motivations, critical thinking, emotional states, and addiction. Grounded in established theories, it offers a foundation for targeted interventions that promote responsible and mindful online behaviors in the digital age.

## Figures and Tables

**Figure 1 ijerph-21-01305-f001:**
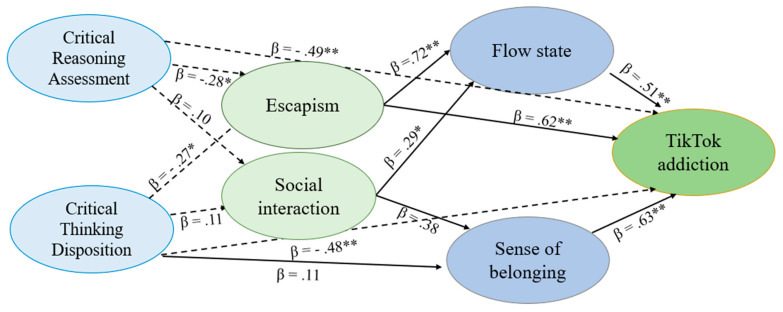
Path diagram analysis with Tiktok addiction as dependent variable. Arrows represent direct relationships. Solid lines indicate positive relationships, while dashed lines indicate negative relationships. Numbers beside the arrows indicate standardized regression coefficients (β). Critical Thinking Disposition (CTD); Cognitive Reflection Ability (CRA).

**Figure 2 ijerph-21-01305-f002:**
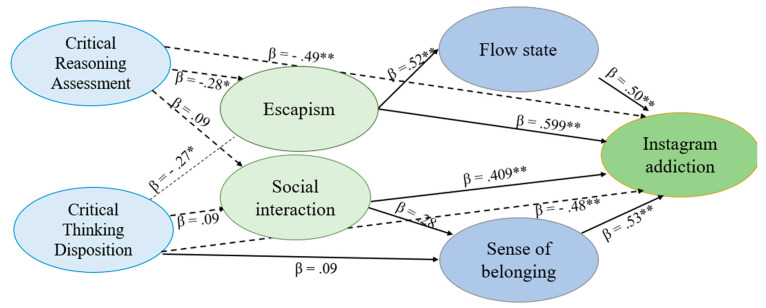
Path diagram analysis with Instagram addiction as dependent variable. Arrows represent direct relationships. Solid lines indicate positive relationships, while dashed lines indicate negative relationships. Numbers beside the arrows indicate standardized regression coefficients (β). Critical Thinking Disposition (CTD); Cognitive Reflection Ability (CRA).

**Table 1 ijerph-21-01305-t001:** Sample sociodemographic profile.

Measure	Item	Frequency (N = 332)	Percentage (%)
Gender	Male	196	58.2
	Female	134	42.6
	LGBT+	2	0.2
Age	Mean	23.53	
	Standard Dev.	3.327	
Education level	Middle School diploma	4	1.5
	High School diploma	158	46.6
	Bachelor’s degree	140	38.7
	Master’s degree	30	13.2
Marital status	Single	304	88
	Cohabiting	24	10
	Divorced	2	1
	Married	2	1

**Table 2 ijerph-21-01305-t002:** Daily hours of use.

Item	Means	Standard Deviation	Min	Max
How many hours a day do you use your smartphone?	7.11	2.99	0	16
How many hours a day do you use TikTok?	0.89	1.54	0	7
How many hours a day do you use Instagram?	2.9	1.53	0	9

**Table 3 ijerph-21-01305-t003:** Summary of descriptive statistics for the investigated measures.

Measure	Mean	Standard Deviation	Min	Max
Escapism (motivation)	16.9	6.7	7	34
Social interaction (motivation)	32.1	10.7	12	55
Flow (experiential state)	41.8	12.8	16	76
Sense of belonging (experiential state)	10.4	5.5	4	27
Critical thinking (disposition)	70.8	9.2	45	88
Total CRA (assessment)	75.63	11.93	47	104
Complexity	11.25	4.96	4	27
Reasoning	13.67	2.74	6	23
Openness	16.75	3.77	6	25
Nature of knowledge	16.05	2.39	11	24
Nature of justification	17.88	2.54	9	24

**Table 4 ijerph-21-01305-t004:** Correlation matrix of social network addiction and motivation, disposition, and state.

	TikTok Addiction	Instagram Addiction	Escapism	Social Interaction	Flow State	Sense of Belonging	CT Disposition	Total CRA
TikTok addiction	-							
Instagram addiction	0.495 **	-						
Escapism	0.593 **	0.579 **	-					
Social interaction	0.284 **	0.402 **	0.378 **	-				
Flow state	0.514 **	0.608 **	0.753 **	0.495 **	-			
Sense of belonging	0.504 **	0.590 **	0.543 **	0.662 **	0.709 **	-		
CT disposition	−0.482 **	−0.511 **	−0.297 **	0.079	−0.274 **	−0.300 **	-	
Total CRA	−0.406 **	−0.399 **	−0.321 **	0.151	−0.264 **	−0.370 **	0.38 **	-

## Data Availability

Data can be obtained through a signed data access agreement. The agreement can be obtained by emailing the contact principal investigator at rafabio@unime.it.

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
