# Peer review of "The Role of Critical Thinking in Mitigating Social Network Addiction: A Study of TikTok and Instagram Users"

_ijerph, 2024, doi:10.3390/ijerph21101305_

Round 1

Reviewer 1 Report

Comments and Suggestions for Authors

Dear Authors, 

Thanks for your work and interesting result, but here are also some points I would like to have considered:

Lines 50-51 mention the dependency on TikTok, which seems unnecessary.

What is the specific role of the full-screen feature?

From lines 70 to 80, it would be better to reconstruct the sentences and the entire paragraph, as there is redundancy with some repeated information.

I am not entirely sure, and I hope the authors are confident because attitudes are not inherently stable traits. Perhaps the current sentence structure conveys a different meaning. Attitudes are stable predictors, as noted by M. Conner et al. (2021), and F. Harreveld et al. (2004) mention that “Attitudes can be based on a stable structure.”

The transition between line 83, where the discussion shifts from attitude to credibility, is not smooth; the authors could consider restructuring it for clarity.

Why focus solely on Instagram and TikTok? Could this not also apply to other social media platforms?

The authors state, “The underlying logic of these hypotheses can be understood through several key pathways. Critical thinkers scrutinize the credibility of online content, reducing susceptibility to misinformation and manipulative SN algorithms.” While this is an interesting claim, it raises the question of why this was not empirically tested among the participants.

 Additionally, the authors assert that “critical thinkers are more aware of their emotional states and motivations for using SNs.” How is this claim supported, particularly if the mediators are factors like boredom or escapism? Does critical thinking still emerge as relevant even when these mediators are present?

Methodology

What is the rationale for restricting the age to 33?

Why is section 2.2 repeated twice?

Which measures assessed the hours spent on social media platforms like Instagram or TikTok?

I am unclear about how there is a positive correlation of 0.590** for the sense of belonging while showing negative correlations of -0.511 and -0.399 with addiction. How do these findings align if users feel a positive sense of belonging and critical reasoning or critical thinking leads to less or more effective use of social networks? Furthermore, if escapism is positively related (0.579) to Instagram addiction, while critical reasoning has a negative correlation (-0.511), how do the authors explain these contradictory correlations? Should we consider that escapism mediates critical reasoning, as I previously suggested?

Also, in Figure 1, how does the sense of belonging relate to critical reasoning ability (CRA) or critical thinking disposition (CTD)?

Regarding the questionnaires, did participants provide responses based on their social network usage over a specific timeframe, such as the last month, the last two weeks, or another period?

Comments on the Quality of English Language

minor edits along with some paragraphs and transitions is needed, 

Author Response

REFEREE 1

Dear Authors,

Thanks for your work and interesting result, but here are also some points I would like to have considered:

Comment 1. Lines 50-51 mention the dependency on TikTok, which seems unnecessary.

Response 1. We agree and delete these lines.

Comment 2. What is the specific role of the full-screen feature?

Response 2. The full-screen feature plays a crucial role in enhancing user engagement by eliminating external distractions and providing a more immersive experience. This feature may contribute to prolonged use of social media platforms, as it intensifies the visual and emotional impact of content, potentially fostering deeper involvement with the platform. In the context of social network addiction, the full-screen mode might encourage users to spend more time interacting with the content, thus playing a role in the addictive patterns of use.

Comment 3. From lines 70 to 80, it would be better to reconstruct the sentences and the entire paragraph, as there is redundancy with some repeated information.

Response 3. Thank you. We rewrote the entire paragraph as follows: The Uses and Gratifications Theory (UGT) provides a framework for understanding problematic social network (SN) use, suggesting that individuals are rational actors seeking to fulfill specific needs through multimedia content consumption (Katz et al., 1973). SN addiction can emerge when external environmental factors influence internal states, potentially leading to addictive behaviors (Kamboj et al., 2018). Users engage with SNs driven by motivations such as seeking social connections, acquiring knowledge, or escapism, which can influence personal dispositions and states experienced during usage. Research identifies several primary motivations for SN use, including the desire for information and inspiration, social interaction, overcoming boredom, and escaping negative emotions (Brailovskaia et al., 2020; Guay, 2022; Han et al., 2022). For instance, using SNs for escapism can lead to dependence, either directly or through the flow state induced by the platform, or by fostering a sense of community belonging (Miranda et al., 2023). Social interaction fulfills the need for connections, creating a "sense of belonging" when individuals feel appreciated by their community (Guo et al., 2016; Meng & Leung, 2021; Chylińska, 2022; Alhabash & Ma, 2017).

Comment 4. I am not entirely sure, and I hope the authors are confident because attitudes are not inherently stable traits. Perhaps the current sentence structure conveys a different meaning. Attitudes are stable predictors, as noted by M. Conner et al. (2021), and F. Harreveld et al. (2004) mention that “Attitudes can be based on a stable structure.”

Response 4. Thank you for pointing out the need for clarification. Our intention was to convey that attitude, including the critical thinking attitude, can be relatively stable over time but not inherently fixed. We recognize that attitudes are predictors of behavior and can be influenced by factors such as cognitive-affective inconsistency and situational variables, as highlighted by Conner et al. (2021) and von Harreveld et al. (2004). We revised the text (highlighted in yellow) to better reflect this understanding and added the suggested literature.

Comment 5. The transition between line 83, where the discussion shifts from attitude to credibility, is not smooth; the authors could consider restructuring it for clarity.

Response 5. Thank you for pointing out the need for a smoother transition between discussing attitudes and credibility. We have restructured the paragraph to clarify the connection between a critical thinking attitude and the challenge of verifying online information. By directly linking critical thinking to the evaluation of information credibility, we aim to emphasize its importance in navigating the complexities of online information, as follows: Attitudes, while not inherently stable traits, serve as consistent predictors of behavior toward SNs over time. For instance, a critical thinking attitude fosters a reflective approach to interpreting online information (Conner et al., 2021; Fabio et al., 2023a; 2023b; von Harreveld, 2004). This reflective approach is particularly crucial given the challenges associated with verifying the reliability of online sources. The lack of expert control on social networks often makes it difficult to assess the credibility of information (Perez et al., 2018; Bürger et al., 2023).

Comment 6. Why focus solely on Instagram and TikTok? Could this not also apply to other social media platforms?

Response 6. Instagram and TikTok are among the most popular social networks in Italy, making them highly relevant for understanding social network addiction in this context. Their unique features, like TikTok's short-form video content and Instagram's visual-centric platform, provide distinct environments for exploring how motivations, attitudes, and states contribute to addictive behaviors. While the findings might apply to other platforms, focusing on these two offers insights into the most prevalent social media experiences in Italy.

Comment 7. The authors state, “The underlying logic of these hypotheses can be understood through several key pathways. Critical thinkers scrutinize the credibility of online content, reducing susceptibility to misinformation and manipulative SN algorithms.” While this is an interesting claim, it raises the question of why this was not empirically tested among the participants.

Response 7. Thank you for your question. While the study aimed to investigate the broader role of critical thinking in mitigating social network addiction, it did not specifically measure the direct scrutiny of online content by participants. Instead, the focus was on assessing general critical thinking attitudes and skills, which are known to contribute to the ability to critically evaluate online information. Future research could include a more direct assessment of participants' behaviors in scrutinizing the credibility of online content to further validate this pathway. We added this in the Limitations and Future Direction section.

Comment 8.  Additionally, the authors assert that “critical thinkers are more aware of their emotional states and motivations for using SNs.” How is this claim supported, particularly if the mediators are factors like boredom or escapism? Does critical thinking still emerge as relevant even when these mediators are present?

Response 8. Thank you. The assertion that “critical thinkers are more aware of their emotional states and motivations for using social networks (SNs)” is supported by existing literature on the relationship between emotional intelligence (EI) and critical thinking. Research has shown that these two constructs are interdependent (Christianson, 2020). Emotional intelligence, defined as the ability to understand and manage one’s own emotions as well as those of others (Mayer, Caruso, & Salovey, 2016), plays a significant role in enhancing critical thinking skills (Yao et al., 2018). Individuals with higher emotional intelligence are better able to process emotional information, which in turn allows them to engage in higher-level critical thinking, particularly when it comes to understanding their motivations and emotional states during SN use. In the context of mediators like boredom and escapism, critical thinking still remains relevant. While boredom is recognized as both a trait and a state emotional condition (Goetz et al., 2010), individuals with higher emotional intelligence are better equipped to manage these emotional states. Goleman (2009) has noted that emotions and thoughts are closely linked, meaning that a person’s ability to regulate emotional responses like boredom could enhance their ability to think critically in such situations. Thus, even when mediators like boredom or escapism are present, individuals with stronger emotional intelligence and critical thinking skills can maintain a level of awareness and self-regulation that mitigates the impact of these mediators. We have added the necessary references at the end of the paragraph in which the assertion is contained (highlighted in yellow).

Methodology

Comment 9. What is the rationale for restricting the age to 33?

Response 9. The decision to restrict the age range to 33 years was informed by several factors. Firstly, the literature indicates that the young adult age group typically extends up to 35 years, as highlighted by the "Committee on Improving the Health, Safety, and Well-Being of Young Adults" in the report Investing in the Health and Well-Being of Young Adults (National Academies Press, 2015). This framework suggests that individuals within this age range share similar developmental, educational, and social experiences, which can influence their behaviors and decision-making processes.

Additionally, our assumption was that the patterns of social network (SN) use would be comparable within this age group. During the recruitment process, we did not receive any responses from individuals aged 34 or 35, which further justified our focus on the 18 to 33 age range. By concentrating on participants within this limit, we aimed to create a relatively homogenous sample, minimizing variability that could arise from age-related differences. This approach enhances the reliability of our findings, allowing us to better explore the relationships between critical thinking, motivations, and social network use.

We have also added this clarification in the limitations and future research section of our study to acknowledge the age restriction and the lack of responses from older participants. Overall, limiting the age range to 33 years aligns with established research on young adulthood and ensures that our study captures relevant trends and behaviors typical of this demographic.

Comment 10. Why is section 2.2 repeated twice?

Response 10. Thank you for bringing this to our attention. The repetition of section 2.2 was an error, and we have corrected it by removing the duplicate section. We appreciate your understanding and oversight in this matter.

Comment 11. Which measures assessed the hours spent on social media platforms like Instagram or TikTok?

Response 11. The measurement of time spent on social media platforms was assessed using a self-report questionnaire. Participants were asked to indicate the amount of time they typically spent on Instagram, TikTok, and smartphones in general on a daily basis, averaging their usage over the past month. This approach was designed to capture an estimate of their recent usage patterns. We added this information on the procedure section (highlighted in yellow).

Comment 12. I am unclear about how there is a positive correlation of 0.590** for the sense of belonging while showing negative correlations of -0.511 and -0.399 with addiction. How do these findings align if users feel a positive sense of belonging and critical reasoning or critical thinking leads to less or more effective use of social networks? Furthermore, if escapism is positively related (0.579) to Instagram addiction, while critical reasoning has a negative correlation (-0.511), how do the authors explain these contradictory correlations? Should we consider that escapism mediates critical reasoning, as I previously suggested?

Response 12.  The correlation of 0.590** for the sense of belonging and the negative correlations of -0.511 and -0.399 with addiction may initially appear contradictory, but these results can be understood by recognizing that different psychological mechanisms underlie these associations. Sense of Belonging and Addiction: The positive correlation between sense of belonging and addiction (r = 0.590**) suggests that individuals who feel a strong sense of belonging through social networks, particularly platforms like Instagram and TikTok, may be more likely to engage in prolonged or excessive use. This sense of belonging could reinforce the desire to stay connected, leading to addictive behaviors, as users seek validation or connection with others in the online space. In this context, the sense of belonging may act as a motivational factor that drives continuous engagement with social media, despite potential negative consequences like addiction.  With reference to Critical Thinking and Addiction, the negative correlations between critical thinking and addiction (r = -0.511 for Instagram, r = -0.399 for TikTok) indicate that individuals with higher critical thinking abilities are less prone to social media addiction. Critical thinkers are more likely to reflect on their motivations for using social networks, regulate their usage, and critically evaluate the content they consume. This reflective process helps them avoid excessive engagement or becoming emotionally dependent on social media for social validation or escapism. With reference to the Interplay of Critical Thinking, Escapism, and Addiction, escapism, as positively correlated with addiction (r = 0.579**), reflects a tendency for individuals to use social networks to avoid real-world stressors. Critical thinking, in contrast, can serve as a protective factor by helping users recognize when they are using social media as an escape mechanism, thereby reducing addictive behaviors. It’s plausible that escapism and critical thinking operate on different psychological levels, where escapism motivates excessive use, while critical thinking mediates or moderates this relationship by curbing excessive engagement.

With reference to your suggestion that escapism may mediate the relationship between critical thinking and addiction is indeed a compelling proposition. While escapism leads to higher levels of social media addiction, individuals with strong critical thinking skills may be more likely to recognize escapist tendencies and regulate their usage accordingly. Thus, escapism could be a mediator, where individuals with low critical thinking are more likely to engage in escapist behaviors, thereby increasing their risk of addiction. We added further mediation analysis to clarify whether escapism serves as an intermediary between critical thinking and addictive behaviors on social networks (highlighted in yellow in the result section).

Comment 13. Also, in Figure 1, how does the sense of belonging relate to critical reasoning ability (CRA) or critical thinking disposition (CTD)?

Response 13. In Figure 1, neither critical reasoning ability (CRA) nor critical thinking disposition (CTD) was found to be directly connected to the sense of belonging. This suggests that the sense of belonging does not have a significant relationship with either CRA or CTD in the model tested.

Comment 14. Regarding the questionnaires, did participants provide responses based on their social network usage over a specific timeframe, such as the last month, the last two weeks, or another period?

Response 14. Yes, the last month; we added this information in the text.

Comments on the Quality of English Language 15. Minor edits along with some paragraphs and transitions is needed.

Response 15. We revised all the text.

Reviewer 2 Report

Comments and Suggestions for Authors

Thank you for the opportunity to review this manuscript. I provide feedback below to assist the manuscript reaching its fullest potential.

1. In the introduction, it would be beneficial to elaborate more on the role of critical thinking with social networks. It seems added on at the end with no explanation for why.

2. I also would like a more declarative statement (or statements) about the benefits of this research. Why is this study important?

3. What did the recruitment announcements say? How many people were contacted? Were the announcements in certain groups or posted on individuals' personal profiles? More details are needed regarding how the study was done, enough where subsequent researchers could replicate the findings. 

4. It looks like there are outliers in the data, as someone reported that on average, they are on their phone 24 hours a day. Even using the +/- 3SD, there appears to be many outliers in the dataset after looking at Table 2. I recommend that the researchers review and remove outliers.

5. Because the data are cross-section, and correlation and regression analyses are used, the author(s) should be careful not to make causal statements, particularly in the discussion section.

6. What are the ranges of the ages? Given the recruitment approach, it's interesting to see that many participants are emerging adults? Was this purposeful? Can the author(s) speak to the demographics of the participants in their study?

Author Response

REFEREE 2

Thank you for the opportunity to review this manuscript. I provide feedback below to assist the manuscript reaching its fullest potential.

Comment 1. In the introduction, it would be beneficial to elaborate more on the role of critical thinking with social networks. It seems added on at the end with no explanation for why.

Response 1. We have revised the introduction section to provide a more thorough explanation of the role of critical thinking in relation to social networks. Specifically, we elaborated on how critical thinking can help individuals assess the content they encounter on social networks, regulate their usage, and resist addictive behaviors. This enhancement aims to highlight the theoretical relevance of critical thinking as a protective factor in the context of social media use.

Comment 2. I also would like a more declarative statement (or statements) about the benefits of this research. Why is this study important?

Response 2. In the revised introduction, we have included a clearer explanation of the importance of this study. By investigating the connection between critical thinking, escapism, and social network addiction, this research contributes to the understanding of how cognitive abilities influence social media usage. The findings may have practical implications for designing interventions to promote healthier social media habits and reduce addiction risks, particularly among young adults.

Comment 3. What did the recruitment announcements say? How many people were contacted? Were the announcements in certain groups or posted on individuals' personal profiles? More details are needed regarding how the study was done, enough where subsequent researchers could replicate the findings.

Response 3.  For the recruitment process, we designed an announcement targeting individuals aged 18 to 35, specifically active users of TikTok and Instagram. The announcement highlighted the study's focus on exploring the relationship between social media use, critical thinking, and motivations for engaging with these platforms. Participants were informed that the survey would take approximately one hour and that certain tests required them not to use any online resources. The announcement also emphasized that no prior knowledge of critical thinking or psychology was necessary, and participation was voluntary and anonymous. A convenience sampling method was employed, where participants were recruited through shared posts in public groups and communities centered around social media usage. The posts were not shared on individual profiles but rather focused on groups relevant to social network research. In total, 400 individuals were contacted, with 332 participants completing the study. The details provided allow for replication, ensuring future studies can follow the same recruitment strategy.

Comment 4. It looks like there are outliers in the data, as someone reported that on average, they are on their phone 24 hours a day. Even using the +/- 3SD, there appears to be many outliers in the dataset after looking at Table 2. I recommend that the researchers review and remove outliers.

Response 4. Thank you for your observation. Upon reviewing the data, we identified one outlier in the phone usage variable and one outlier in the TikTok usage variable (as you can see in the data set). We applied a standard outlier detection method using a +/- 3 standard deviations (SD) criterion, as suggested. Both outliers were subsequently removed from the dataset. After removing these outliers, we re-ran the analysis, and the revised results are reflected in the updated Table 2. This ensures that the findings are more robust and not unduly influenced by extreme data points.

Comment 5. Because the data are cross-section, and correlation and regression analyses are used, the author(s) should be careful not to make causal statements, particularly in the discussion section.

Response 5. Thank you for your observation. We acknowledge the limitations of our cross-sectional design and have taken care to revise the discussion section to avoid any causal statements. Throughout the revised discussion, we have emphasized that the findings represent correlations rather than causal relationships. We also highlighted the need for longitudinal studies to better assess the directionality of the relationships observed in our analysis.

Comment 6. What are the ranges of the ages? Given the recruitment approach, it's interesting to see that many participants are emerging adults? Was this purposeful? Can the author(s) speak to the demographics of the participants in their study?

Response 6. Thank you for your question. The age range of participants in our study was set between 18 and 33 years, with a mean age of 23.53 years (SD = 3.32). The decision to target young adults was purposeful and grounded in the literature, which typically defines young adulthood as extending up to 35 years (Bonnie et al., 2015). This age group was chosen to ensure homogeneity in the sample, as young adults tend to share similar social media usage patterns, critical thinking development stages, and motivations for using platforms like TikTok and Instagram.Our recruitment approach through social media platforms such as WhatsApp, Facebook, and Instagram naturally attracted a high percentage of emerging adults, but the focus on young adults was intentional to align with the study's objectives. We acknowledge this demographic focus and have included it in the discussion of our study’s design and sample characteristics.

Round 2

Reviewer 2 Report

Comments and Suggestions for Authors

I appreciate the authors' thorough responses and edits based on my previous comments. I believe the manuscript has significantly improved. 

Comments on the Quality of English Language

There were some areas where the writing was confusing, particularly when aligning with singular and plural nouns.